# Antibiotic knowledge, attitudes and practices: new insights from cross-sectional rural health behaviour surveys in low-income and middle-income South-East Asia

Marco J Haenssgen,[1,2,3,4] Nutcha Charoenboon,[5] Giacomo Zanello,[6,7] Mayfong Mayxay,[8,9,10] Felix Reed-Tsochas,[3,4,11] Yoel Lubell,[1,5] Heiman Wertheim,[12,13] Jeffrey Lienert,[3,4,14] Thipphaphone Xayavong,[15,16] Yuzana Khine Zaw,[17] Amphayvone Thepkhamkong,[8] Nicksan Sithongdeng,[8] Nid Khamsoukthavong,[8] Chanthasone Phanthavong,[8] Somsanith Boualaiseng,[8] Souksakhone Vongsavang,[8] Kanokporn Wibunjak,[5] Poowadon Chai-in,[5] Patthanan Thavethanutthanawin,[5] Thomas Althaus,[1,5] Rachel Claire Greer,[5,18] Supalert Nedsuwan,[19] Tri Wangrangsimakul,[18,20] Direk Limmathurotsakul,[20] Elizabeth Elliott,[21,22] Proochista Ariana[1]

For numbered affiliations see end of article.

**Correspondence to**
Dr Marco J Haenssgen;
marco.haenssgen@warwick.ac.uk

## ABSTRACT

**Introduction** Low-income and middle-income countries (LMICs) are crucial in the global response to antimicrobial resistance (AMR), but diverse health systems, healthcare practices and cultural conceptions of medicine can complicate global education and awareness-raising campaigns. Social research can help understand LMIC contexts but remains under-represented in AMR research.

**Objective** To (1) Describe antibiotic-related knowledge, attitudes and practices of the general population in two LMICs. (2) Assess the role of antibiotic-related knowledge and attitudes on antibiotic access from different types of healthcare providers.

**Design** Observational study: cross-sectional rural health behaviour survey, representative of the population level.

**Setting** General rural population in Chiang Rai (Thailand) and Salavan (Lao PDR), surveyed between November 2017 and May 2018.

**Participants** 2141 adult members (≥18 years) of the general rural population, representing 712 000 villagers.

**Outcome measures** Antibiotic-related knowledge, attitudes and practices across sites and healthcare access channels.

**Findings** Villagers were aware of antibiotics (Chiang Rai: 95.7%; Salavan: 86.4%; p<0.001) and drug resistance (Chiang Rai: 74.8%; Salavan: 62.5%; p<0.001), but the usage of technical concepts for antibiotics was dwarfed by local expressions like 'anti-inflammatory medicine' in Chiang Rai (87.6%; 95% CI 84.9% to 90.0%) and 'ampi' in Salavan (75.6%; 95% CI 71.4% to 79.4%). Multivariate linear regression suggested that attitudes against over-the-counter antibiotics were linked to 0.12 additional antibiotic use episodes from public healthcare providers in Chiang Rai (95% CI 0.01 to 0.23) and 0.53 in Salavan (95% CI 0.16 to 0.90).

## Strength and limitations of this study

► Provincial-level representative survey using a three-stage stratified cluster random sampling design.
► Survey based on preceding qualitative research on antibiotic use in South-East Asia.
► Inclusion of the general population enables insights into formal and informal healthcare utilisation.
► Cross-sectional analysis of rural health behaviours excludes seasonal change and urban settings.
► Two-month recall period enabled greater inclusion but may bias responses towards better educated population groups.

**Conclusions** Locally specific conceptions and counterintuitive practices around antimicrobials can complicate AMR communication efforts and entail unforeseen consequences. Overcoming 'knowledge deficits' alone will therefore be insufficient for global AMR behaviour change. We call for an expansion of behavioural AMR strategies towards 'AMR-sensitive interventions' that address context-specific upstream drivers of antimicrobial use (eg, unemployment insurance) and complement education and awareness campaigns.

**Trial registration number** Clinicaltrials.gov identifier NCT03241316.

## INTRODUCTION

Antimicrobial resistance (AMR) threatens modern medicine by rendering antimicrobial drugs ineffective. Multifaceted global strategies target human, animal and plant health alongside the environment, food production

and safety to respond to this 'superbug crisis'.[1] In human health, supply side responses include incentives to stimulate drug research and development; action on the demand side intends to limit and target antimicrobial use, for instance, through new diagnostic technologies, public health interventions to improve vaccine coverage and hygiene, and other antimicrobial stewardship activities like restricted dispensing of antibiotics and prescriber feedback.[2–4] As an interdisciplinary field, the social dimensions of the problem are being recognised in global AMR policy, which are typically addressed via education and awareness-raising activities aimed at governmental staff, healthcare workers and the general public.[2]

Low-income and middle-income countries (LMICs) play an important role in the global response to AMR. However, diverse health systems, healthcare practices and conceptions related to the use of antimicrobials require social research to understand local contexts and the complexity of human behaviour in LMIC settings. For example, with a focus on the health behaviour of the general public, the anthropological literature suggests that social factors like precarity and discrimination can influence medicine use independently of awareness;[5] psychology and behavioural economics indicate that health decision-making processes interact with the social environment and contextual change to create adverse behavioural biases;[6 7] and communication studies research points at interferences between awareness campaigns and local contexts that can entail unforeseen consequences like politicisation, stigmatisation or accidentally encouraging the behaviours they try to discourage.[8 9] Such examples underline the possible contribution of the social sciences to AMR, but they remain persistently under-represented with less than 2% of all AMR-related publications (see online supplementary file 1 - figure A1 for a time trend).[10] This under-representation is problematic for at least three reasons:

► We currently have an insufficient social science knowledge base for behavioural interventions in AMR—a global health priority that has attracted more than £600 million of expenditure and future commitments for research and surveillance.[11–13]
► The recent withdrawal of large pharmaceutical companies from antimicrobial research and development[14] threatens the AMR supply side response, requiring yet more effective action on the demand side.
► More extensive social sciences work can yield novel social innovations as a benefit of disciplinary diversification.[15]

Using social research methods, the objectives of this paper were to (1) Describe antibiotic-related knowledge, attitudes and practices of the general population in two LMICs. (2) Assess the role of antibiotic-related knowledge and attitudes on antibiotic access from different types of healthcare providers. We report findings from a provincial-level representative survey of rural health behaviours across 69 villages in northern Thailand and 65 villages in southern Lao PDR as part of the interdisciplinary 'Antibiotics and Activity Spaces' project.[16] (This paper contributes to the project's research question, *'What are the manifestations and determinants of problematic antibiotic use in patients' healthcare-seeking pathways?'.*[16]) We implemented the study in South-East Asia, which is characterised as a region 'at high risk of the emergence and spread of antibiotic resistance in humans'.[17 18] With more than 9% of global air passengers and more than 110 million international tourist arrivals in 2016,[19] the potential of cross-border spread of drug-resistant microbes also gives AMR research in South-East Asia a global relevance—as the recent importation of multidrug-resistant *Neisseria gonorrhoeae* to the UK showed.[20] Within South-East Asia, Thailand and Lao PDR lent themselves for a comparative analysis because of their physical and cultural proximity, and Chiang Rai (Thailand) and Salavan (Lao PDR) in particular had similar terrain and large and ethnically diverse rural populations. The main field site differences were Thailand's more advanced economic and health system context and more established AMR action plan.[21] For example, Thailand maintains a national strategic plan on AMR (2017–2021),[22] its public health expenditure per capita in 2016 was nearly 10 times higher than Lao's (US$496.2 vs US$50.1 in purchasing power parity), and Thailand had 2.3 nurses per 1000 people in 2015, compared with 1.0 per 1000 people in Lao PDR in 2014.[19]

## METHODS
### Multistage survey design
Our study design was a three-stage stratified cluster random survey (figure 1): Following the purposive selection of five districts per province, we selected a random sample of 30 primary sampling units (PSUs) per province (six per district), stratified by distance to the nearest district headquarters. The second stage was the selection of an interval sample of 5% but at least 30 of all households in the PSU, which we approximated as residential structures on satellite maps.[23] Participants were sampled in the third and final stage. This process involved the random selection of available household members (one for every five members). At each sampling stage, we substituted unavailable selections (1) With a stratified random replacement for the random PSU sample. (2) With the nearest available neighbour for the interval sample of households. (3) With a simple random replacement for the random household member sample (replacement numbers indicated in figure 1). The cross-sectional data collection took place between November 2017 and May 2018.

### Study population
Our study population was the general adult population of rural Chiang Rai and Salavan (522000 in Chiang Rai and 190000 in Salavan as per census data), from whom we drew a representative sample of 1158 villagers in Chiang Rai and 983 in Salavan. We did not specifically sample patients, but we recorded any acute illness episode or

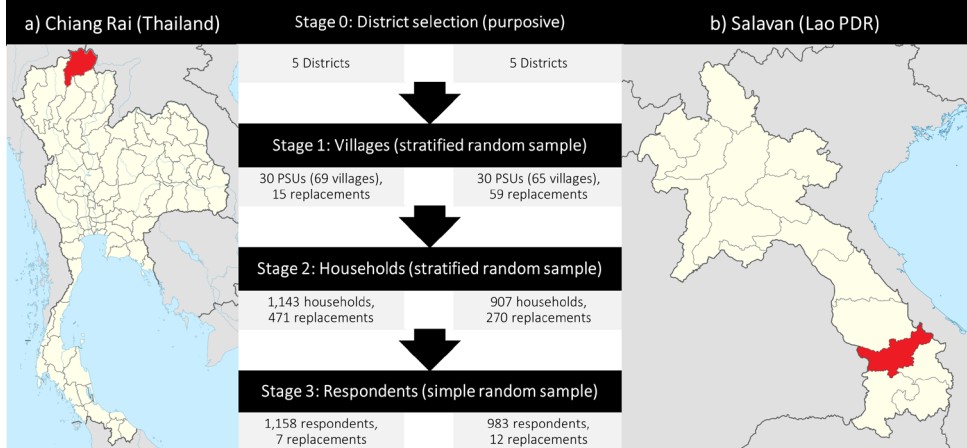

**Figure 1** Survey sites and multistage sampling process (adapted from Wikimedia Commons [36]). Notes: Unavailable selections at each sampling stage were substituted with a random replacement for the random samples of PSUs and household members, and with the nearest available neighbour for the interval sample of households. One PSU could contain more than one administrative village; if the first-chosen village contained less than 600 houses, then adjacent villages would be included. PSU, primary sampling unit.

accident-related injury if one occurred within the last 2 months of the interview, both for the respondents and any children under their supervision.

## Patient and public involvement

This study did not sample patients but only adult members of the general public. The survey instrument was based on preceding qualitative research in South-East Asia,[24 25] in which patients, healthcare providers and healthy adults participated, but patients or members of the public were not directly involved in the design or conception of the study. This preceding research prompted the research interest in treatment-seeking behaviours and conceptions of medicine and illness among the broader rural population in South-East Asia. We will disseminate our findings through outreach to policy stakeholders and local development organisations, through public engagement activities like the World Antibiotic Awareness Week, and though our local network of collaborators in the field sites.

## Data collection

Our survey instrument was a 45 min face-to-face questionnaire (see online supplementary file 2). It was administered on tablets running the survey software SurveyCTO (Dobility, Cambridge, Massachusetts, USA) by locally recruited survey teams comprising seven enumerators and two survey supervisors per country, who received 5 days of full-time classroom and field training. The original English questionnaire was co-developed with, and translated into Thai and Lao by, the local research team (we refrained from additional back-translation as the local language versions of the questionnaire were based on qualitative research material that we had previously used in the region, aided further by field pilots and cognitive interviewing), and local translators were recruited for the 228 instances where we encountered language barriers. The questionnaires were piloted in

rural Chiang Rai and Salavan, with 50 cognitive interviews supporting the questionnaire development and revision as well as the contextualisation of the survey data (not reported here; interview guide in online supplementary file 3).[26]

The questionnaire covered basic demographic and socioeconomic information, antibiotic-related knowledge and attitudes, and treatment-seeking behaviour during acute illnesses and accident-related injuries. When measuring people's awareness of antibiotics, we could not simply ask villagers whether they knew what 'antibiotics' are, considering that (1) A variety of local terms related to antibiotics existed. (2) People may be familiar with specific antibiotic brands but not aware of their antibiotic attributes. (3) The understanding of technical language was uncommon (see Results section for evidence on this point). We therefore asked respondents first if they recognised images of common antibiotics in the field site (three images on the survey tablet in Chiang Rai and a bag with seven local antibiotics in Salavan, considering the wider range of terms and medicines in circulation in the Lao site; see questionnaire in online supplementary file 2). In the 108/1974 (5.5%) cases where the respondents did not mention 'antibiotics', its colloquial equivalents, or the names of specific antibiotic types, we asked them if they had heard about 'anti-inflammatory drugs' ('ยาแก้อักเสบ' or *'yah kae ak seb'*) in Thai and 'germ resisters' ('ยาต້ານເຊື້ອ' or *'yah dtan suea'*) in Lao as common local notions of 'antibiotics'. We next asked about the purposes for which the respondent would use these antibiotics, which served as information alongside inputs from local pharmacists to triangulate in later parts of the questionnaire whether the respondent received antibiotics during an illness. However, 752/2986 (25.2%) medicine use episodes could not be confirmed as either antibiotic or non-antibiotic (eg, 'white powder' or 'green capsule'). We included these uncertain cases as

'potential' antibiotic use episodes to capture behaviour more comprehensively.

## Data analysis

In order to inform the current global health agenda on antibiotic education and awareness raising, we used descriptive statistical analysis and regression analysis to describe the patterns of knowledge and attitudes—and their role in determining antibiotic use—across the two field sites, using the variables described in online supplementary file 1 - table A1. If the common policy narrative holds, then we would expect rural populations in Chiang Rai and Salavan to exhibit:

► Low degrees of antibiotic-related knowledge.
► Generally high levels of antibiotic consumption especially from informal sources (eg, unregistered shops selling antibiotics over the counter).
► Lower general antibiotic use and a higher share of supervised antibiotic use from formal healthcare providers among people whose attitudes correspond to awareness-raising messages for AMR (based on FAO/OIE/WHO material, see online supplementary file 1 - table A1).[1]

We stratified and compared the samples by field site (ie, province) to account for the systemic influence of the health system configuration on people's health knowledge and behaviour. We estimated provincially representative patterns using poststratification weights based on census data (considering village size and district-specific age and gender composition) and adjusting the descriptive and regression analysis results for the multistage sampling design with the help of the svy suite of commands in Stata V.15 (StataCorp, College Station, Texas, USA). We separately analysed the full sample and the subset of respondents who reported a recent illness, whereby we tested differences in knowledge, attitudes and behaviours across provinces and across antibiotic access channels with $X^2$ tests for binary and Wilcoxon rank-sum tests (two-sided) for non-normally distributed variables. We further carried out multivariate analysis of the determinants of antibiotic use from public (public hospitals and primary care units), private (private hospitals, clinics and pharmacies) and informal sources (grocery stores selling medicine, traditional healers). The multivariate analysis used linear regression models adjusted by the complex survey design (sampling clusters and survey weights), which we compared across the two country samples, using the Chow test to ascertain systematic differences in the determinants of antibiotic use across the two field sites.[27]

Although the dependent variables were not normally distributed, the otherwise preferable functional form of Poisson regression did not converge in most cases owing to the relatively small sample sizes. However, where they did converge, the linear regressions yielded more conservative estimates. Likewise, the linear regressions adjusted by the complex survey design yielded more conservative results than linear multilevel models that take the hierarchical structure of the data into account. We therefore

present the linear regression results in this article. For improved model fitness and to reduce the influence of outliers, we further substituted the duration of the illness with its log. To test for multicollinearity in the cross-sectional survey data, we analysed the pairwise correlations between all independent variables stratified by field site, whereby the largest correlation coefficients in Chiang Rai were +0.59 (ethnicity/religion) and –0.50 (education/age), and in Salavan +0.76 (ethnicity/religion) and +0.62 (religion/wealth) (see online supplementary file 1 - table A3). The largest variance inflation factors (VIFs) were for the dummy variables of religion (VIF = 3.12 in the Salavan sample) and ethnicity (VIF = 2.01 in the Chiang Rai sample), the exclusion of which from the regression models did not produce meaningful differences in parameter estimates or significance levels of the other independent variables. We therefore presented the full regression models to not omit independent variables selectively. We indicated significance levels below 0.1, 0.05 and 0.01 with *, ** and ***, respectively.

## RESULTS

Representative statistical data of Chiang Rai and Salavan are presented in online supplementary appendix table 1. In terms of sociodemographic characteristics of the rural population in the two provinces, Chiang Rai villagers were older on average (p<0.001), tended to have received more formal education (p<0.001) and had higher asset wealth (p<0.001), while fewer Salavan villagers belonged to the local majority ethnicity (p=0.030).

Respondents' recognition of antibiotics in Chiang Rai was significantly higher than in Salavan, but overall high in both sites (Chiang Rai: 95.7%; Salavan: 86.4%; p<0.001). Recognition of the phrase 'drug resistance' was high as well, whereby 74.8% recognised the term in Chiang Rai and 62.5% (p<0.001) recognised either of the two common variations in Salavan. Online supplementary appendix table 1 further indicates that antibiotic-related knowledge and attitudes aligned more closely with FAO/OIE/WHO (Food and Agriculture Organization of the United Nations, World Organisation for Animal Health, World Health Organization) messages in Chiang Rai than in Salavan (p<0.001 for all four questions). Across the four questions, respondents in rural Chiang Rai had an average answer score of 1.8 as opposed to rural Salavan with 0.7 (p<0.001).

Figure 2, Panel A, demonstrates the ways in which people related to 'antibiotics'. In Chiang Rai, respondents commonly referred to antibiotics as 'anti-inflammatory drug', representing 87.6% of all responses ('ยาแก้อักเสบ' or '*yah kae ak seb*'; a vernacular notion specific to antibiotics). (Actual anti-inflammatory medicine like ibuprofen would usually be referred to by its brand names.) Only 7.2% used the official term for 'antibiotic' ('ยาปฏิชีวนะ' or '*yah pa ti chee wa na*') alongside '*germ killer*' and specific antibiotic types like '*corlam*' (chloramphenicol; 4.6%). In rural Salavan, a larger portion of 38.6% used the official

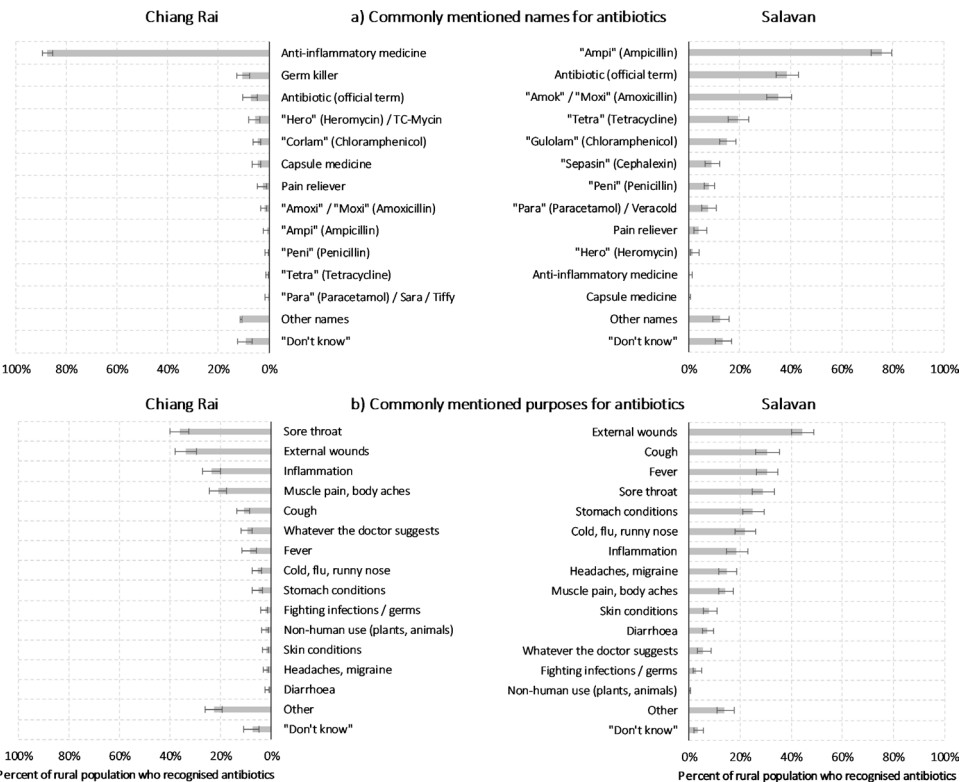

**Figure 2** Common names and purposes for antibiotics. Source: Authors' analysis of survey data. Notes: Only including respondents who indicated that they had seen the presented medicine (ie, common antibiotics) before. Chiang Rai: n=1076; Salavan: n=775. Population-weighted statistics, accounting for complex survey design. Multiple response permitted. Error bars indicate 95% CI.

term for antibiotics ('ยาต້ານເຊື້ອ' or '*yah dtan suea'*, translated as 'germ resister, but Salavan respondents were also more likely to use various colloquial expressions for specific types of antibiotics, like '*Ampi'* with 75.6% and '*Amok'* with 35.3%.

Reported purposes of antibiotic use were yet more varied and are displayed in figure 2, Panel B. The overall most common use was the treatment of external wounds (Chiang Rai: 33.7%; Salavan: 44.4%; p<0.001). Other frequently reported uses in Salavan included coughs (30.5%; Chiang Rai: 10.9%; p<0.001) and fevers (30.5%; Chiang Rai: 8.3%; p<0.001). Thai respondents further indicated common use of antibiotics for sore throats (Chiang Rai: 36.3%, where it was the single most common use; Salavan: 28.9%; p=0.016) and for the more general idea of an 'inflammation' of the body (Chiang Rai: 23.5%; Salavan: 18.6%; p=0.083). Thai respondents would also more often limit their use to whatever a healthcare worker would recommend (Chiang Rai: 9.5%; Salavan: 5.4%; p=0.037), while 2.3% indicated that they would treat their plants or animals (dogs and chickens) with antibiotics (Salavan: 0.1%; p<0.001). Using antibiotics to treat infections or to fight bacteria and germs was only mentioned by a small minority of the rural populations (Chiang Rai: 2.4%; Salavan: 2.8%; p=0.243).

Table 1 indicates that—though people typically recognised the term 'drug resistance'—the responses to the question *'What do you think is drug resistance?'* only

rarely corresponded to clinical definitions, and the coexistence of two common translations of the term in Lao PDR complicated the picture further. In Chiang Rai, 10.6% of the interpretations related to antibiotics and/or drug-resistant germs. Lao respondents linked the official term *'due yah'* to clinical definitions in 7.7% of all interpretations, and the colloquial term *'lueng yah'* in 9.6% of all interpretations. Not unlike other high-income and low-income countries,[28] drug resistance was typically interpreted as a growing tolerance of the body towards medicine as a result of repeated use (not limited to antibiotics). Other common interpretations in Chiang Rai were the incorrect or erratic use of medicine (12.5%), and an understanding of drug resistance as side effects of or allergic reactions to medicine in general (4.2%). In Salavan, *'due yah'* was often interpreted as a refusal or 'stubbornness' to take medicine (21.8%; possibly due to its literal translation into *'stubborn (to the effect of) medicine'*), while its vernacular equivalent *'lueng yah'* was often interpreted in the opposite way as a psychological dependence or addiction to medicine (24.9%).

According to our survey data, 99.9% of the rural population in Chiang Rai and 91.6% of the rural Salavan population had a public primary healthcare centre within a 10 km radius. Private sources were more varied, as 93.0% and 34% of the rural Chiang Rai population had a private clinic and a pharmacy within a 10 km radius, respectively (37.8% and 47.4% in Salavan, respectively), whereas

**Table 1** Awareness and interpretations of 'drug resistance'

| | Chiang Rai | | Salavan | | |
|---|---|---|---|---|---|
| | '*due yah*' | | '*due yah*' | | '*lueng yah*' |
| Awareness in rural population | 72.9% (67.4–77.8) | | 27.1% (22.2–32.6) | | 58.8% (54.6–63.0) |
| Top five interpretations | | | | | |
| Rank 1 | Body becomes tolerant to medicine | 54.1% (49.3–58.9) | Body becomes tolerant to medicine | 38.1% (30.4–46.4) | Body becomes tolerant to medicine | 50.9% (44.7–57.1) |
| Rank 2 | Taking medicine incorrectly | 12.5% (10.2–15.3) | Patient is 'stubborn', refuses medicine | 21.8% (14.8–31.0) | Addicted to/preference for medicine | 24.9% (20.2–30.2) |
| Rank 3 | **Reference to antibiotics, drug-resistant germs** | **10.6% (8.1–13.8)** | Side effects, drug allergy | 9.2% (5.2–15.8) | **Reference to antibiotics, drug-resistant germs** | **9.6% (6.9–13.1)** |
| Rank 4 | Don't know | 6.3% (4.6–8.7) | **Reference to antibiotics, drug-resistant germs** | **7.7% (4.7–12.5)** | Don't know | 4.0% (2.3–6.8) |
| Rank 5 | Side effects, drug allergy | 4.2% (2.6–6.7) | Addicted to/preference for medicine | 7.1% (3.5–13.8) | Sickness is 'stubborn'/ unresponsive | 2.9% (1.3–6.2) |

Source: Authors' analysis of survey data.
Notes: Ranking percentages only include respondents who indicated that they had heard the respective term 'drug resistance' before. 95% CIs in parentheses. Chiang Rai: n=871; Salavan (*due yah*): n=206; Salavan (*lueng yah*): n=470. Population-weighted statistics, accounting for complex survey design. Only single response permitted. In Salavan, the common response '*due yah*' means *lueng yah*' (24.8% (18.4–32.6)) was recoded to incorporate respondent's definition of *lueng yah*. Entries in **bold** relate directly to clinical definition of antimicrobial resistance.

informal healthcare through shops and informal healers was nearly universally available within the survey villages (>97.8% in all cases).

Among our 2141 respondents, we captured 608 illness episodes in Chiang Rai and 356 in Salavan (see online supplementary appendix table 1). Healthcare utilisation during these episodes varied slightly across the two field sites. Chiang Rai respondents accessed a narrower spectrum of healthcare providers and were significantly less likely to access public and 'other' healthcare providers (p<0.001 in both cases). Both sites also exhibited a high level of medicine access, with 2.2 and 2.5 medicine use episodes during an illness in Chiang Rai and Salavan, respectively (p=0.050). Respondents in Chiang Rai thereby indicated higher use of non-antibiotic medicine (Chiang Rai: 1.6; Salavan: 1.3; p=0.048). In contrast, respondents in Salavan had more episodes of antibiotic use per illness (Chiang Rai: 0.2; Salavan: 0.4; p<0.001), and more usage of medicines that could potentially include antibiotics (Chiang Rai: 0.4; Salavan: 0.9; p<0.001). The pattern of antibiotic access was similar for informal sources, but generally lower in Chiang Rai: confirmed antibiotic use from informal channels represented 1.6% of all medicine use episodes in Chiang Rai and 3.3% in Salavan; and 3.6% in Chiang Rai and 7.9% in Salavan if unconfirmed but potential antibiotic use episodes are included.

Comparing the bivariate differences between individuals who accessed antibiotics from public, private and informal sources (see online supplementary file 1 - table A2) indicated that, contrary to intuition, patients receiving antibiotics from informal sources had no less wealth or formal education than users of public healthcare. Indeed, wealthier and more educated individuals in Chiang Rai were significantly associated with receiving antibiotics from informal sources (wealth: p=0.012; education: p=0.032). Similarly, awareness of drug resistance was not significantly lower among patients who received antibiotics from informal sources, while the share of respondents who linked drug resistance to biomedical notions of AMR in Salavan was significantly higher among individuals accessing antibiotics through informal channels compared with public channels (13.4% vs 4.4%, p=0.030). Patients who accessed antibiotics through informal channels were nevertheless significantly more inclined towards buying over-the-counter antibiotics than public antibiotic users (Chiang Rai: p=0.040; Salavan: p<0.001).

The results of the multivariate analysis are presented in table 2. The dependent variables were the number of confirmed antibiotic use episodes from public, private and informal sources; as a sensitivity check, we also included the more inclusive definition of 'confirmed and potential' antibiotic use episodes. The relationship between antibiotic-related knowledge and attitudes and antibiotic use was mixed. Among the most common knowledge-related and attitude-related predictors of antibiotic use was the respondents' inclination to buy antibiotics. Again contrary to expectations, the attitude to not buy antibiotics over the counter was linked to disproportionate

consumption of antibiotics from public sources. For example, patients who would not buy antibiotics over the counter would have 0.12 additional potential antibiotic use episodes in Chiang Rai (Model 7; 95% CI 0.01 to 0.23) and up to 0.53 in Salavan (Model 8; 95% CI 0.16 to 0.90), ceteris paribus. Also knowledge of antibiotics was positively associated with antibiotic consumption from private sources in Chiang Rai (Model 3) and from public and informal sources in Salavan (Models 2 and 6). In contrast, other antibiotic-related attitudes and knowledge linked negatively to antibiotic consumption. For example, the knowledge that antibiotic resistance can spread was linked to 0.25 fewer potential antibiotic use episodes from informal sources (Model 12, 95% CI −0.39 to −0.10) and Chiang Rai patients who preferred alternatives over antibiotics had 0.12 fewer confirmed antibiotic use episodes from private sources (Model 3; 95% CI −0.22 to −0.03).

Among other covariates, wealthier patients had lower consumption of antibiotics from public (Chiang Rai and Salavan, Models 7 and 8) and higher consumption from private (Salavan, Model 4) and informal healthcare providers (Chiang Rai, Models 5 and 11)—presumably enabled by their higher purchasing power. Speakers of the majority language were also more likely to consume more antibiotics from public sources, which resonates with anecdotes encountered during the field research according to which speakers of minority languages tended to be less assertive in the patient-doctor encounter (the link to education was less clear). Not surprisingly, longer illness episodes were also associated with more antibiotic use episodes from public and private sources.

The Chow test indicated that the determinants of confirmed antibiotic use from private sources were significantly different in Chiang Rai and Salavan (Models 3 and 4; p=0.011), as were all sources of potential and confirmed antibiotic use (Models 7 to 12; p=0.056, p=0.015 and p=0.083 for public, private and informal antibiotic use, respectively).

## DISCUSSION AND CONCLUSION

Our paper aimed at understanding AMR-related general population behaviour in LMICs through a description of antibiotic-related knowledge, attitudes and practices in rural Thailand and Lao PDR, and an assessment of the role of antibiotic-related knowledge and attitudes on antibiotic access from different types of healthcare providers. We demonstrated that rural populations exhibited:

► Mixed but surprisingly high levels of awareness and attitudes corresponding to AMR awareness-raising material, although only a minority of villagers were familiar with technical notions of antibiotics and drug resistance.
► Relatively low levels of antibiotic access from informal sources.
► Surprisingly counterintuitive links between informal antibiotic use, socioeconomic status, and their

**Table 2** Regression results of determinants of antibiotic use from public, private and informal sources

| Site | Confirmed antibiotic use episodes | | | | | | Confirmed and potential antibiotic use episodes | | | | | |
|---|---|---|---|---|---|---|---|---|---|---|---|---|
| | Public providers | | Private providers | | Informal providers | | Public providers | | Private providers | | Informal providers | |
| Model number | CR (1) | SAL (2) | CR (3) | SAL (4) | CR (5) | SAL (6) | CR (7) | SAL (8) | CR (9) | SAL (10) | CR (11) | SAL (12) |
| **Independent variables** | | | | | | | | | | | | |
| Would not buy antibiotics over the counter | 0.045* (−0.01 to 0.10) | 0.168** (0.02 to 0.32) | 0.074 (−0.02 to 0.16) | −0.040 (−0.13 to 0.05) | −0.050 (−0.11 to 0.01) | −0.008 (−0.09 to 0.08) | 0.117** (0.01 to 0.23) | 0.530*** (0.16 to 0.90) | 0.151* (−0.03 to 0.33) | −0.095 (−0.33 to 0.14) | −0.031 (−0.10 to 0.04) | 0.021 (−0.14 to 0.18) |
| Prefers antibiotics over alternatives | −0.023 (−0.08 to 0.03) | −0.017 (−0.12 to 0.08) | −0.124** (−0.22 to −0.03) | −0.009 (−0.10 to 0.08) | 0.033 (−0.02 to 0.08) | −0.014 (−0.09 to 0.07) | −0.026 (−0.17 to 0.12) | −0.222* (−0.44 to 0.00) | −0.161* (−0.35 to 0.03) | 0.054 (−0.18 to 0.28) | 0.005 (−0.07 to 0.08) | 0.044 (−0.10 to 0.19) |
| Does not keep antibiotics for future use | −0.049* (−0.10 to 0.00) | −0.008 (−0.11 to 0.09) | −0.060 (−0.14 to 0.02) | −0.013 (−0.12 to 0.09) | −0.024 (−0.08 to 0.03) | 0.037 (−0.07 to 0.15) | −0.073 (−0.20 to 0.05) | 0.185 (−0.18 to 0.55) | −0.056 (−0.25 to 0.14) | 0.056 (−0.26 to 0.37) | −0.049 (−0.12 to 0.02) | −0.050 (−0.21 to 0.11) |
| Knows that antibiotic resistance can spread | 0.041 (−0.05 to 0.13) | −0.088 (−0.35 to 0.17) | 0.030 (−0.11 to 0.17) | −0.051 (−0.34 to 0.24) | 0.035 (−0.04 to 0.11) | −0.155*** (−0.25 to −0.06) | 0.068 (−0.15 to 0.29) | −0.355 (−1.53 to 0.82) | −0.060 (−0.25 to 0.13) | −0.138 (−0.73 to 0.46) | 0.013 (−0.09 to 0.11) | −0.247*** (−0.39 to −0.10) |
| Aware of antibiotics | −0.028 (−0.12 to 0.07) | 0.101* (−0.00 to 0.20) | 0.109** (0.01 to 0.21) | 0.017 (−0.08 to 0.11) | −0.135 (−0.38 to 0.10) | 0.064** (0.01 to 0.12) | 0.093 (−0.20 to 0.39) | 0.185 (−0.28 to 0.65) | −0.197 (−0.78 to 0.39) | −0.346 (−0.94 to 0.25) | −0.073 (−0.33 to 0.18) | 0.130* (−0.00 to 0.26) |
| Aware of drug resistance† | −0.041 (−0.11 to 0.03) | 0.056 (−0.08 to 0.20) | 0.056 (−0.03 to 0.14) | 0.086 (−0.04 to 0.21) | 0.016 (−0.01 to 0.05) | 0.014 (−0.09 to 0.12) | −0.143* (−0.29 to 0.01) | −0.041 (−0.39 to 0.31) | −0.063 (−0.33 to 0.20) | 0.113 (−0.16 to 0.38) | −0.024 (−0.09 to 0.04) | −0.027 (−0.20 to 0.14) |
| Female | −0.005 (−0.06 to 0.06) | 0.086* (−0.01 to 0.18) | −0.005 (−0.09 to 0.08) | −0.064 (−0.17 to 0.04) | 0.035 (−0.02 to 0.09) | 0.049 (−0.03 to 0.13) | −0.122 (−0.27 to 0.03) | 0.168 (−0.09 to 0.42) | 0.001 (−0.19 to 0.20) | −0.296* (−0.59 to 0.00) | 0.028 (−0.04 to 0.09) | 0.181** (0.03 to 0.33) |
| Age | −0.001 (−0.00 to 0.00) | −0.001 (−0.00 to 0.00) | 0.003* (−0.00 to 0.00) | −0.001 (−0.00 to 0.01) | −0.002* (−0.00 to 0.00) | 0.002 (−0.00 to 0.00) | 0.001 (−0.00 to 0.01) | −0.001 (−0.01 to 0.01) | 0.009** (0.00 to 0.02) | −0.004 (−0.01 to 0.00) | −0.001 (−0.00 to 0.00) | 0.007** (0.00 to 0.01) |
| Education (years) | −0.002 (−0.01 to 0.01) | −0.012* (−0.02 to 0.00) | 0.004 (−0.01 to 0.02) | 0.012 (−0.00 to 0.03) | −0.006*** (−0.01 to −0.00) | 0.010 (−0.00 to 0.03) | 0.001 (−0.01 to 0.02) | −0.001 (−0.05 to 0.05) | 0.015 (−0.01 to 0.04) | 0.018 (−0.01 to 0.05) | −0.001 (−0.01 to 0.01) | 0.010 (−0.01 to 0.03) |
| Speaking Thai/Lao | 0.161*** (0.07 to 0.25) | 0.251*** (0.10 to 0.40) | 0.005 (−0.12 to 0.13) | −0.010 (−0.12 to 0.10) | 0.019 (−0.02 to 0.06) | 0.058 (−0.01 to 0.13) | −0.016 (−0.46 to 0.43) | 0.161 (−0.27 to 0.59) | −0.336 (−0.79 to 0.12) | −0.005 (−0.30 to 0.29) | 0.033 (−0.05 to 0.12) | 0.152** (0.01 to 0.30) |
| Wealth Index | −0.179 (−0.42 to 0.06) | −0.105 (−0.47 to 0.26) | −0.001 (−0.28 to 0.28) | 0.340* (−0.01 to 0.69) | 0.158** (0.01 to 0.31) | −0.121 (−0.35 to 0.10) | −0.417*(−0.90 to 0.06) | −1.073* (−2.26 to 0.12) | 0.323 (−0.30 to 0.95) | 0.505 (−0.30 to 1.31) | 0.241** (0.02 to 0.46) | 0.065 (−0.73 to 0.86) |
| Buddhist religion | −0.016 (−0.11 to 0.08) | −0.065 (−0.29 to 0.16) | −0.074 (−0.21 to 0.06) | −0.031 (−0.18 to 0.12) | 0.010 (−0.02 to 0.04) | 0.028 (−0.06 to 0.12) | −0.033 (−0.22 to 0.16) | 0.037 (−0.40 to 0.47) | −0.122 (−0.44 to 0.19) | −0.031 (−0.35 to 0.29) | 0.017 (−0.05 to 0.08) | −0.021 (−0.23 to 0.19) |
| Thai/Lao nationality | −0.030 (−0.13 to 0.07) | −0.140 (−0.53 to 0.25) | −0.069 (−0.24 to 0.10) | 0.013 (−0.11 to 0.13) | −0.003 (−0.06 to 0.06) | 0.088 (−0.03 to 0.21) | 0.009 (−0.37 to 0.39) | 0.003 (−0.43 to 0.44) | 0.005 (−0.52 to 0.53) | −0.118 (−0.46 to 0.22) | −0.078 (−0.24 to 0.09) | −0.603 (−1.95 to 0.74) |
| Majority ethnic group (Thai/Lao Loum) | 0.033 (−0.04 to 0.11) | 0.169* (−0.02 to 0.36) | 0.065 (−0.04 to 0.17) | −0.059 (−0.19 to 0.07) | 0.050** (0.01 to 0.09) | −0.039 (−0.14 to 0.06) | 0.049 (−0.11 to 0.21) | 0.360* (−0.04 to 0.76) | −0.044 (−0.27 to 0.18) | 0.118 (−0.20 to 0.44) | 0.035 (−0.03 to 0.10) | 0.030 (−0.17 to 0.23) |
| Self-rated severity (1=mild, 2=medium, 3=severe) | 0.063*** (0.02 to 0.10) | 0.009 (−0.08 to 0.09) | 0.002 (−0.05 to 0.05) | 0.062 (−0.02 to 0.14) | 0.009 (−0.02 to 0.04) | 0.023 (−0.03 to 0.08) | 0.077 (−0.02 to 0.17) | 0.254** (0.01 to 0.50) | 0.068 (−0.06 to 0.20) | 0.107 (−0.17 to 0.39) | 0.065** (0.01 to 0.12) | −0.038 (−0.20 to 0.12) |
| Log of duration of illness episode (days) | 0.060*** (0.02 to 0.10) | 0.036 (−0.01 to 0.08) | 0.076*** (0.03 to 0.12) | 0.046 (−0.01 to 0.10) | 0.003 (−0.02 to 0.02) | −0.025 (−0.07 to 0.02) | 0.275*** (0.16 to 0.39) | 0.507*** (0.27 to 0.74) | 0.257*** (0.10 to 0.41) | 0.234** (0.05 to 0.42) | 0.005 (−0.03 to 0.04) | −0.012 (−0.13 to 0.10) |
| Constant | −0.007 (−0.23 to 0.21) | −0.123 (−0.55 to 0.30) | −0.152 (−0.38 to 0.08) | −0.198 (−0.46 to 0.07) | 0.117 (−0.12 to 0.35) | −0.207* (−0.43 to 0.02) | −0.022 (−0.61 to 0.56) | −0.767 (−1.82 to 0.29) | −0.181 (−1.08 to 0.72) | 0.029 (−1.17 to 1.23) | −0.006 (−0.37 to 0.35) | 0.199 (−1.18 to 1.58) |

Continued

**Table 2** Continued

| Site | Confirmed antibiotic use episodes | | | | | | Confirmed and potential antibiotic use episodes | | | | | |
|---|---|---|---|---|---|---|---|---|---|---|---|---|
| | Public providers | | Private providers | | Informal providers | | Public providers | | Private providers | | Informal providers | |
| Model number | CR (1) | SAL (2) | CR (3) | SAL (4) | CR (5) | SAL (6) | CR (7) | SAL (8) | CR (9) | SAL (10) | CR (11) | SAL (12) |
| **Model statistics** | | | | | | | | | | | | |
| Number | 604 | 356 | 604 | 356 | 604 | 356 | 604 | 356 | 604 | 356 | 604 | 356 |
| $R^2$ | 0.102 | 0.106 | 0.080 | 0.133 | 0.104 | 0.046 | 0.143 | 0.218 | 0.126 | 0.136 | 0.052 | 0.095 |
| F statistic | 2.29*** | 3.34*** | 1.89** | 1.79** | 0.89 | 1.48 | 2.82*** | 4.57*** | 2.41*** | 1.30 | 1.18 | 2.27*** |
| Chow test | (2)–(1)=0 | | (4)–(3)=0 | | (6)–(5)=0 | | (8)–(7)=0 | | (10)–(9)=0 | | (12)–(11)=0 | |
| F statistic | 1.33 | | 2.05** | | 1.08 | | 1.65* | | 1.97** | * | 1.55* | |

Source: Authors' analysis of survey data.

Notes: Illness episode level, including completed illnesses experienced by respondent or child under their supervision, excluding incomplete episodes. Population-weighted statistics, accounting for complex survey design. 95% CIs in parentheses.

*p<0.1, **p<0.05, ***p<0.01.

†Comparing Thai 'due yah' with the combined Lao 'due yah' and 'lueng yah'.

CR, Chiang Rai; SAL, Salavan.

attitudes—especially among villagers in Salavan, who had disproportionately high antibiotic use if their attitude showed a disinclination against over-the-counter antibiotics.

Our survey data also revealed profound differences between the two field sites despite their cultural and geographical proximity. For example, villagers referred to antibiotics with wide-ranging and locally specific vernacular expressions (only a minority adopted technical language in either site), and 'drug resistance' was typically understood as a general tolerance of the body to medicine but local interpretations ranged from patients refusing medicine to patients being addicted to medicine. Some of these differences could be explained by the local health system configuration. The better endowed and more regulated health system as well as the more extensive public health campaigns in Chiang Rai arguably contributed to the higher rates of public awareness and the comparatively lower rates of antibiotic use, whereas the Salavan health system faced more pressing trade-offs between ensuring access to and preventing the overuse of antibiotics. Alas, as the analysis has shown, antibiotic-related awareness and attitudes appeared to have little bearing on people's antibiotic consumption when controlling for other determinants of medicine use.

The surveys were implemented after the monsoon season to reduce accessibility barriers like landslides, floods and farm work. This temporal focus meant that our survey was not able to capture internal migration or seasonal change affecting the epidemiological environment. The rural survey is also unable to speak for urban health behaviour or behavioural patterns outside rural Thailand and Lao PDR, or for awareness and behaviour among healthcare staff and policy makers (with which awareness-raising activities for the general public may interact). Lastly, our focus on health behaviour and our 60-day recall period could introduce recall and social desirability biases. Most LMIC health behaviour research uses recall periods of 14–30 days; longer recall periods can lead to under-representation of lower educated groups.[29] However, for a survey of behaviour rather than of epidemiological patterns, 14-day recall would have truncated the sample to an impractical size (omitting 540/964 (56.0%) of all responses) and neglected that illness episodes often extended beyond a fortnight (as was the case for 91/964 (8.7%) of the recorded illnesses). In response, we conducted regular review sessions with our survey team to identify and alleviate social desirability; we excluded chronic illnesses; and our questionnaire asked our respondents to walk through the sequence of events, which improves recall.[30] While we cannot rule out a residual risk of social desirability and recall bias, it is not clear a priori whether and how any remaining bias would affect our comparison of antibiotic uses across different healthcare providers.

By virtue of being a representative rural survey in northern Thailand and southern Lao PDR, the specific notions and behavioural patterns around antibiotic use

are not generalisable beyond the study context, although similar interpretations of antibiotics as 'anti-inflammatory medicine' exist elsewhere (eg, in China),[31] and even the documented practice of using antibiotics in plant cultivation has historical antecedents.[32] Other study findings like the widespread use of antibiotics for external (and often allegedly 'internal') wounds have few documented equivalents in other settings and deserve further research. However, the findings of our study have a broader relevance insofar as they expose the complexity of local knowledge and its relationship to AMR-related behaviour. On the one hand, our work underlines the challenges facing public awareness campaigns as the current principal strategy to change AMR-related population behaviour. For example, if not mindful of the local context, the slogan of the 2017 World Antibiotic Awareness Week to 'use antibiotics wisely to combat rising drug resistance' could plausibly entail *increased* antibiotic use or the use of stronger medicine if people understand drug resistance as stubbornness of patients or as a problem applying to all types of medicines.

On the other hand, our study also demonstrated that the link between knowledge, attitudes and antibiotic-related behaviour may be weak in LMIC contexts. This disjunction is not new,[5] but the counterintuitive relationship between awareness, antibiotic-related attitudes and antibiotic use from informal sources suggests that AMR-related information can easily entail unintended consequences—knowledge and awareness empower, but people themselves decide how they will use this new 'power' in their daily lives.[33] For instance, villagers may not necessarily buy antibiotics from private clinics and unregulated corner shops because of ignorance, but because they become more assertive about their health (and increased wealth may enable patients to exercise this assertiveness).

Considering potential misunderstandings in AMR communication on the one hand and contextual determinants of behaviour beyond knowledge deficits on the other, we call for an expansion of behavioural AMR strategies to address structural factors of behavioural change. For example, vulnerability and adversity may drive people into seemingly irrational antimicrobial use.[34] A sick labourer or factory worker may take antimicrobials desperately to maintain their job and to sustain their families, in which case it would be futile trying to convince them that their hardship is secondary to the global health goal of tackling AMR. Yet, it may be possible to alleviate their pressure to consume antimicrobials through paid sick leave and unemployment insurance. We propose the exploration of such 'AMR-sensitive interventions' to address upstream drivers of antimicrobial use and to complement education and awareness campaigns—similar to nutrition-sensitive interventions that target the determinants of malnutrition and undernutrition through upstream interventions like social safety nets (rather than, eg, providing supplements directly to people).[35] AMR-sensitive interventions require us to venture out of health policy terrain into broader development policy. There is yet little evidence whether and how such context-oriented approaches bear fruit. Greater involvement of the social sciences is necessary to uncover this gap and to find constructive solutions that address the social factors of which AMR is a symptom.

**Author affiliations**
[1]Center for Tropical Medicine and Global Health, University of Oxford Centre for Tropical Medicine, Oxford, UK
[2]School of Cross Faculty Studies, University of Warwick, Coventry, UK
[3]Green Templeton College, Oxford, United Kingdom
[4]Said Business School, University of Oxford, Oxford, UK
[5]Mathematical/Economic Modelling, Mahidol Oxford Tropical Medicine Research Unit, Bangkok, Thailand
[6]School of Agriculture, Policy and Development, University of Reading, Reading, UK
[7]Leverhulme Centre for Integrative Research on Agriculture and Health, London, , UK
[8]Lao-Oxford-Mahosot Hospital-Wellcome Trust Research Unit, Mahidol Oxford Tropical Medicine Research Unit, Vientiane, Laos
[9]Faculty of Postgraduate Studies, University of Health Sciences, Vientiane, Laos
[10]Institute of Research and Education Development, University of Health Sciences, Vientiane, Laos
[11]Oxford Martin School, University of Oxford, Oxford, UK
[12]Medical Microbiology Department, Radboud University, Nijmegen, The Netherlands
[13]Oxford University Clinical Research Unit, Ho Chi Minh City, Viet Nam
[14]National Human Genome Research Institute, National Institutes of Health, Bethesda, Maryland, USA
[15]Department of Peace and Conflict Studies, University for Peace, Ciudad Colon, Costa Rica
[16]Department of Political Science, Ateneo de Manila University, Quezon City, Philippines
[17]Department of Global Health and Development, London School of Hygiene and Tropical Medicine, London, UK
[18]Chiangrai Clinical Research Unit, Chiangrai Regional Hospital, Chiang Rai, Thailand
[19]Primary Care Department, Chiangrai Regional Hospital, Chiang Rai, Thailand
[20]Microbiology, Mahidol Oxford Tropical Medicine Research Unit, Bangkok, Thailand
[21]UCL Anthropology, University College London, London, UK
[22]Institut de recherche pour le développement, Vientiane, Laos

**Twitter** https://twitter.com/HaenssgenJ

**Acknowledgements** The authors thank their collaborators Caroline OH Jones, Romyen Kosaikanont, Pollavat Praphattong, Pathompong Manohan, Paul N Newton, Sommay Keomany, Penporn Warapikuptanun, Daranee Intralawan, Patchapoom U-Thong, Patipat Benjaroon, Narinnira Sangkham, Sirirat Chailert, Krittanon Promsutt, Maipheth Keovilayvanh and Phaengnitta Phanthasomchit for guidance during the study design and for field survey implementation assistance. The authors also thank Ernest R Guevarra and Eric Ohuma for helpful discussions, feedback and support during the survey design phase; Edward Gibbs, Lisa White, the Centre for Tropical Medicine and Global Health, the CABDyN Complexity Centre, and the Mahidol-Oxford Tropical Medicine Research Unit/Wellcome Trust Strategic Award for institutional support; Zoë Doran and the Clinical Trials Support Group at the Mahidol Oxford Tropical Medicine Research Unit for quality assurance during the protocol development process; Alison Brindle, Claire-Lise Kessler and John Bleho for engaging communication work; and Karen Valentine, Holly Blades and Kanchana Pongsaswat for indispensable operational and logistical assistance.

**Contributors** Study conceptualisation, design and theoretical framing: MJH. Study design: MJH, NC, GZ, MM, FR-T, YL, HFLW, JL, TX, YKZ, AT, NS, NK, CP, SB, SV, KW, PC-I, PT, TA, RCG, SN, TW, DL, EE, PA. Survey instrument development: MJH, GZ, NC. Study protocol development: MJH, NC. Data cleaning and coding: MJH, NC. Data analysis and manuscript draft: MJH. Manuscript review and approval: MJH, NC, GZ, MM, FR-T, YL, HFLW, JL, TX, YKZ, AT, NS, NK, CP, SB, SV, KW, PC-I, PT, TA, RCG, SN, TW, DL, EE, PA.

**Funding** This project is funded by the Antimicrobial Resistance Cross Council Initiative supported by the seven research councils in partnership with the Department of Health and Department for Environment Food & Rural Affairs (grant ref. ES/P00511X/1, administered by the UK Economic and Social Research Council). YKZ was further supported by internal research placement funding by the MSc International Health and Tropical Medicine (University of Oxford).

**Disclaimer** The funders had no involvement in the design and implementation of the project.

**Map disclaimer** The depiction of boundaries on the map(s) in this article do not imply the expression of any opinion whatsoever on the part of BMJ (or any member of its group) concerning the legal status of any country, territory, jurisdiction or area or of its authorities. The map(s) are provided without any warranty of any kind, either express or implied.

**Competing interests** None declared.

**Patient consent for publication** Not required.

**Ethics approval** The research was reviewed and approved by the University of Oxford Tropical Research Ethics Committee (Ref. OxTREC 528-17), and it received local ethical approval in Thailand from the Mae Fah Luang University Research Ethics Committee on Human Research (Ref. REH 60099), and in Lao PDR from the National Ethics Committee for Health Research (Ref. NEHCR 074). Participation in the survey was voluntary and we obtained informed verbal consent from all participants, which was audio recorded and documented by the survey field investigators with a written record of oral consent for each participant.

**Provenance and peer review** Not commissioned; externally peer reviewed.

**Data availability statement** Data are available in a public, open access repository.

**Open access** This is an open access article distributed in accordance with the Creative Commons Attribution 4.0 Unported (CC BY 4.0) license, which permits others to copy, redistribute, remix, transform and build upon this work for any purpose, provided the original work is properly cited, a link to the licence is given, and indication of whether changes were made. See: https://creativecommons.org/licenses/by/4.0/.

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
