## [Reviewer comments · BMJ Open]

ARTICLE DETAILS

TITLE (PROVISIONAL)	Antibiotic Knowledge, Attitudes, and Practices: New Insights from Cross-Sectional Rural Health Behaviour Surveys in Low- and Middle-Income Southeast Asia
AUTHORS	Haenssngen, Marco; Charoenboon, Nutch; Zanello, Giacomo; Mayxay, Mayfong; Reed-Tsochas, Felix; Lubell, Yoel; Wertheim, Heiman; Lienert, Jeffrey; Xayavong, Thippaphone; Khine Zaw, Yuzana; Thepkhamkong, Amphayvone; Sithongdeng, Nicksan; Khamsoukthavong, Nid; Phanthavong, Chanthasone; Boualaiseng, Somsanith; Vongsavang, Souksakhone; Wibunjak, Kanokporn; Chai-in, Poowadon; Thavethanutthanawin, Patthanant; Althaus, Thomas; Greer, Rachel; Nedsuwan, Supalert; Wangrangsimakul, Tri; Limmathurotsakul, Direk; Elliott, Elizabeth; Ariana, Proochista

VERSION 1 – REVIEW

REVIEWER	Esmita Charani Imperial College London United Kingdom
REVIEW RETURNED	06-Jan-2019

GENERAL COMMENTS	Thank you for inviting me to review this manuscript. This is an important addition to the literature in the field of AMR particularly in LMICs. This is a very interesting study with findings that are useful in providing insights into knowledge and attitudes and practices in antibiotic use in community settings. It is not clear how the participants were sampled, from reading this current manuscript. Could the authors please provide a little detail on this in the methods. There is not data in the methods and analysis employed or the findings that this is a 'social sciences' study. Rather it is a social research survey. According to the manuscript: 'The questionnaire covered basic demographic and socio-economic information, antibiotic-related , knowledge and attitudes, and treatment-seeking behaviour during acute illnesses and accident-related injuries.' The questionnaire was completed in a face-to-face 45-min session. Can the authors provide what social science methods. There is no qualitative data presented and no indication that the questionnaires included in-depth questions or qualitative exploration of the findings. I would therefore suggest to the authors to change the narrative and not present this study, which is important and useful as a survey on antibiotic use and knowledge and perceptions from LMICs, as a social sciences study. The findings of this research can be used to develop and
---

	undertake an in-depth qualitative research exploring the contextual drivers for some of the reported behaviours. Perhaps in the discussion it will be of interest to have discussed the findings around the association of increased wealth with access to antibiotics from informal sources. Could the profound differences between the two sites also be considered from the perspective of the different level of public health campaigns around AMR in these different settings? What about the differences in healthcare providers in community (formal and informal) in these different settings. What is the distribution of pharmacy outlets in these two different settings? Or unaccredited suppliers of medicines? The other interesting finding was that antibiotics were most frequently used for external skin wounds - this is different to other studies of antibiotic use in the community. This would be an interesting follow up study to identify the reasons e.g. increased RTAs, poor surgical care, animal bites.
--	--

REVIEWER	M Gualano University of Torino, Italy
REVIEW RETURNED	22-Jan-2019

GENERAL COMMENTS	I think the manuscript is suitable for publication in its current form
--

REVIEWER	Timo Lajunen NTNU, Norway
REVIEW RETURNED	12-Apr-2019

GENERAL COMMENTS	The paper presents a study about antibiotic knowledge, attitudes, and practices among the rural population in Lao PDR and Thailand. The topic is definitely important, and the paper is quite well written. I am not sure about the novelty value, though. Still, I think that information about antibiotic use in various countries Comments  1. The paper suffers from the same lack of focus as many public health studies. The objective of the paper “was to contribute to the understanding of LMIC contexts and AMR-related human behavior from a social science perspective”. So, the aim is to describe. Why not have a more ambitious aim to explain the AMR-related human behavior with the variables included in the study? One reason for the lack of focus is this lack of specified objective. 2. The paper is based on the comparison of two villages. Why the comparison of these two villages is important? If the reason is that the Thai village “more advanced economic and health system context”, the authors should quantify and specify the difference. They should use this information in the analysis. 3. The questionnaire was translated to the local languages. Was the back translation method used to evaluate the translation? 4. The statistical treatment is insufficient. X2 tests for binary and Wilcoxon rank-sum tests (two-sided) were used. Even with non-parametric variables, more developed statistical techniques could
---

	have been used (e.g. multiway frequency analysis). In fact, this data could have been analyzed even with regression models. In that case, however, the authors should have decided what the focus actually was. For example, they could have tested two different regression models explaining AMR-behavior in the villages included. That would have shown similarities and differences between villages. 5. The statistics are not reported sufficiently. Instead of reporting p-values, you need to report the test statistics (e.g. chi-square value, Z-test score, etc). P-value is nothing but the test of significance and should be reported by using significance stars. In fact, the p-value is not enough and also 95%CI should be reported. 6. There are far too many tables and appendixes. These should be given as added material, not part of the manuscript. In sum, the paper and data are interesting but the lack of focus makes it difficult to follow the paper.
--	--

VERSION 1 – AUTHOR RESPONSE

Reviewers' Comments to Author:

Reviewer: 1

Reviewer Name: Esmita Charani

Institution and Country: Imperial College London, United Kingdom

Please state any competing interests or state 'None declared': None to declare.

Thank you for inviting me to review this manuscript. This is an important addition to the literature in the field of AMR particularly in LMICs. This is a very interesting study with findings that are useful in providing insights into knowledge and attitudes and practices in antibiotic use in community settings.  We thank the reviewer for this assessment.

It is not clear how the participants were sampled, from reading this current manuscript. Could the authors please provide a little detail on this in the methods.

 The manuscript described the three-stage survey research design and participant sampling in the first paragraph of the methods section and in Figure 1. For clarification, we have highlighted that, "Participants were sampled in the third and final stage. This process involved was the random selection of available household members (one for every five members)."

There is not data in the methods and analysis employed or the findings that this is a 'social sciences' study. Rather it is a social research survey.

According to the manuscript: 'The questionnaire covered basic demographic and socio-economic information, antibiotic-related, knowledge and attitudes, and treatment-seeking behaviour during acute illnesses and accident-related injuries.' The questionnaire was completed in a face-to-face 45-min session. Can the authors provide what social science methods. There is no qualitative data presented and no indication that the questionnaires included in-depth questions or qualitative exploration of the findings. I would therefore suggest to the authors to change the narrative and not present this study, which is important and useful as a survey on antibiotic use and knowledge and perceptions from LMICs, as a social sciences study. The findings of this research can be used to develop and undertake an in-depth qualitative research exploring the contextual drivers for some of the reported behaviours.

 We argue that surveys have been a standard tool in the social sciences and that social sciences research is not limited to qualitative research. In this particular case, the quantitative survey followed previous qualitative research for an explanatory study led by the field of development studies with inputs from sociology, economics, social anthropology, and public health and tropical medicine. However, we concede that this manuscript does not primarily address a social science audience (as can be seen e.g. in the absence of a theoretical framework), owing to which we have removed the wording “social science perspective” from the objective.

Perhaps in the discussion it will be of interest to have discussed the findings around the association of increased wealth with access to antibiotics from informal sources.

 While space is limited to expand on the point of wealth and health behaviour, we expanded the presentation of the results relating to wealth (“Among other covariates, wealthier patients had lower consumption of antibiotics from public (Chiang Rai and Salavan, Models 7 and 8) but also higher consumption from private (Salavan, Model 4) and informal healthcare providers (Chiang Rai, Models 5 and 11) – presumably enabled by their higher purchasing power”) and added another short in the discussion section (“increased wealth may enable patients to exercise this assertiveness”).

Could the profound differences between the two sites also be considered from the perspective of the different level of public health campaigns around AMR in these different settings? What about the differences in healthcare providers in community (formal and informal) in these different settings. What is the distribution of pharmacy outlets in these two different settings? Or unaccredited suppliers of medicines?

 To add more information on the local healthcare landscapes, we added a paragraph in the results section stating that, “According to our survey data, 99.9% of the rural population in Chiang Rai and 91.6% of the rural Salavan population had a public primary healthcare centre within a 10km radius. Private sources were more varied, as 93.0% and 34% of the rural Chiang Rai population had a private clinic and a pharmacy within a 10km radius, respectively (37.8% and 47.4% in Salavan, respectively), whereas informal healthcare through shops and informal healers was nearly universally available within the survey villages (>97% in all cases).”

 In the discussion, we added further explanation to situate the findings in the health system context. Specifically, we added the following text: “Some of these differences could be explained by the local health system configuration. The better endowed and more regulated health system as well as the more extensive public health campaigns in Chiang Rai arguably contributed to the higher rates of public awareness and the comparatively lower rates of antibiotic use, whereas the Salavan health system faced more pressing trade-offs between ensuring access to and preventing the overuse of antibiotics. Alas, as the analysis has shown, antibiotic-related awareness and attitudes appeared to have little bearing on people’s antibiotic consumption when controlling for other determinants of medicine use.”

The other interesting finding was that antibiotics were most frequently used for external skin wounds - this is different to other studies of antibiotic use in the community. This would be an interesting follow up study to identify the reasons e.g. increased RTAs, poor surgical care, animal bites.

 We thank the reviewer for highlighting this – we have added a sentence in the discussion to bring this point to the fore: “Other study findings like the widespread use of antibiotics for external (and often allegedly “internal”) wounds have few documented equivalents in other settings and deserve further research.”

Reviewer: 2

Reviewer Name: M Gualano

Institution and Country: University of Torino, Italy

Please state any competing interests or state 'None declared': none declared

I think the manuscript is suitable for publication in its current form.

 We thank the reviewer for this assessment.

Reviewer: 3

Reviewer Name: Timo Lajunen

Institution and Country: NTNU, Norway

Please state any competing interests or state 'None declared': None declared

The paper presents a study about antibiotic knowledge, attitudes, and practices among the rural population in Lao PDR and Thailand. The topic is definitely important, and the paper is quite well written. I am not sure about the novelty value, though. Still, I think that information about antibiotic use in various countries

Comments

1. The paper suffers from the same lack of focus as many public health studies. The objective of the paper "was to contribute to the understanding of LMIC contexts and AMR-related human behavior from a social science perspective". So, the aim is to describe. Why not have a more ambitious aim to explain the AMR-related human behavior with the variables included in the study? One reason for the lack of focus is this lack of specified objective.

 We agree with the reviewer's assessment and have refined the stated objective: "The objective of this paper was to (1) describe antibiotic-related knowledge, attitudes, and practices of the general population in two LMICs and to (2) assess the role of antibiotic-related knowledge and attitudes on antibiotic access from different types of healthcare providers."

2. The paper is based on the comparison of two villages. Why the comparison of these two villages is important? If the reason is that the Thai village "more advanced economic and health system context", the authors should quantify and specify the difference. They should use this information in the analysis.

 For clarification, the research took place in 134 villages (60 primary sampling units) rather than two villages. The survey data was representative on the provincial level. We have made this point more explicit in the introduction to avoid misunderstanding, but we feel that the final introductory paragraph provides already justification why we compare Chiang Rai and Salavan as two provinces in Southeast Asia.

 We also added more detail on the different health system context, both in the introduction (where we now state that, "For example, Thailand maintains a national strategic plan on antimicrobial resistance (2017-2021). In addition, according to World Bank data, Thailand's public health expenditure per capita in 2016 were nearly ten times higher than Lao's (USD 496.2 vs. USD 50.1 in purchasing power parity), and Thailand had 2.3 nurses per 1,000 people in 2015, compared to 1.0 per 1,000 people in Lao PDR in 2014.") and in the results (where we explain which part of the rural population is covered by which health facilities).

 Because the health system context applies to all villages in each site similarly, it cannot be added as an explanatory variable. However, the sample stratification (2 site-specific sub-samples) was intended to take account of the systemic influence of the health system configuration on behaviour – in line with common practice in the social sciences. We made this point more explicit in the data analysis section of the methodology.

 In the discussion, we situate the findings again in the health system context. Specifically, we added the following explanations: "Some of these differences could be explained by the local health system configuration. The better endowed and more regulated health system as well as the more extensive public health campaigns in Chiang Rai arguably contributed to the higher rates of public awareness and the comparatively lower rates of antibiotic use, whereas the Salavan health system faces more pressing trade-offs between ensuring access to and preventing the overuse of antibiotics. Alas, as the analysis has shown, antibiotic-related awareness and attitudes have little bearing on people's antibiotic use when controlling for other determinants of medicine use."

3. The questionnaire was translated to the local languages. Was the back translation method used to evaluate the translation?

 The questionnaires were developed jointly by the tri-lingual survey team – first in English and then translated into Thai and Lao. The translations were unproblematic as the research team had carried out preliminary qualitative research locally to prepare this survey, the meaning of each survey item was discussed with the local fieldworker teams (aided by a comprehensive survey manual), and the local-language questionnaires were tested through pilot surveys and cognitive interviewing. In light of the local embeddedness of the research team and the extensive testing of the survey materials, an additional round of formal back-translation was not deemed necessary. For explanation, we added a footnote that states, "We refrained from additional back-translation as the local-language versions of the questionnaire were based on qualitative research material that we had previously used in the region, aided further by field pilots and cognitive interviewing."

4. The statistical treatment is insufficient. X2 tests for binary and Wilcoxon rank-sum tests (two-sided) were used. Even with non-parametric variables, more developed statistical techniques could have been used (e.g. multiway frequency analysis). In fact, this data could have been analyzed even with regression models. In that case, however, the authors should have decided what the focus actually was. For example, they could have tested two different regression models explaining AMR-behavior in the villages included. That would have shown similarities and differences between villages.

 In light of the revised objective (following the suggestion of the reviewer), we have included regression models to explain antibiotic use from different sources in Chiang Rai and Salavan. The multivariate analysis used linear regression models adjusted by the complex survey design (sampling clusters and survey weights), which we compared across the two country samples, using the Chow test to ascertain systematic differences in the determinants of antibiotic use across the two field sites. We added this explanation to the methodology and report the findings at the end of the results section.

 As the inclusion of a regression model requires further statistical consideration, we included the following footnote in the methodology section: "Although the dependent variables were not normally distributed, the otherwise preferable functional form of Poisson regression did not converge in most cases owing to the relatively small sample sizes. However, where they did converge, the linear regressions yielded more conservative estimates (likewise, the linear regressions adjusted by the complex survey design yielded more conservative results than linear multilevel models that take the hierarchical structure of the data into account). We therefore present the linear regression results in this article. For improved model fitness and to reduce the influence of outliers, we further substituted the duration of the illness with its log. To test for multicollinearity in the cross-sectional survey data, we analysed the pairwise correlations between all independent variables stratified by field site, whereby the largest correlation coefficients in Chiang Rai were +0.59 (ethnicity/religion) and -0.50 (education/age), and in Salavan +0.76 (ethnicity/religion) and +0.62 (religion/wealth) (see Table A3 in the Supplemental Material). The largest variance inflation factors (VIFs) were for the dummy variables of religion (VIF = 3.12 in the Salavan sample) and ethnicity (VIF = 2.01 in the Chiang Rai sample), the exclusion of which from the regression models did not produce meaningful differences in parameter estimates or significance levels of the other independent variables. We therefore presented the full regression models to not omit independent variables selectively."

5. The statistics are not reported sufficiently. Instead of reporting p-values, you need to report the test statistics (e.g. chi-square value, Z-test score, etc). P-value is nothing but the test of significance and should be reported by using significance stars. In fact, the p-value is not enough and also 95%CI should be reported.

 We updated the table of summary statistics and the supplementary table A3 (comparing characteristics of individuals who received antibiotics from public, private, and informal sources) with X²-values/z-scores and included significance stars as suggested. For the new table reporting the regression results, we reported significance stars and 95% confidence intervals for all regressors, and significance stars and F-statistic for the model tests and Chow tests.

 For in-text descriptions of differences between sites, we prefer to continue to rely on p-values for legibility and in accordance with other BMJ Open publications focusing on antibiotic use (e.g. Greer et al., BMJ Open, 8:e022250, 2018).

6. There are far too many tables and appendixes. These should be given as added material, not part of the manuscript.

 We have moved the table of summary statistics to the appendix, removed Figure 3 (bivariate comparison of antibiotic sources), and moved all previous appendix tables and figures to the supplemental material. The main manuscript thus contains 2 figures and 2 tables, all of which we deem essential for understanding methodology and results.

In sum, the paper and data are interesting but the lack of focus makes it difficult to follow the paper.